# A Null *B*-Ring Improves the Antioxidant Levels of Flavonol: A Comparative Study between Galangin and 3,5,7-Trihydroxychromone

**DOI:** 10.3390/molecules23123083

**Published:** 2018-11-26

**Authors:** Xiaojian Ouyang, Xican Li, Wenbiao Lu, Xiaojun Zhao, Dongfeng Chen

**Affiliations:** 1School of Chinese Herbal Medicine, Guangzhou University of Chinese Medicine, Guangzhou 510006, China; oyxiaojian55@163.com (X.O.); luwb1@gzucm.edu.cn (W.L.); zxj@gzucm.edu.cn (X.Z.); 2Innovative Research & Development Laboratory of TCM, Guangzhou University of Chinese Medicine, Guangzhou 510006, China; 3School of Basic Medical Science, Guangzhou University of Chinese Medicine, Guangzhou 510006, China; 4The Research Center of Basic Integrative Medicine, Guangzhou University of Chinese Medicine, Guangzhou 510006, China

**Keywords:** antioxidant, galangin, 3,5,7-trihydroxychromone, flavonol, structure–activity, *B*-ring

## Abstract

To clarify the role of the *B*-ring in antioxidant flavonols, we performed a comparative study between galangin with a null *B*-ring and 3,5,7-trihydroxychromone without a *B*-ring using five spectrophotometric assays, namely, ^•^O_2_^−^-scavenging, 1,1-diphenyl-2-picrylhydrazyl radical (DPPH^•^)-scavenging, 2-phenyl-4,4,5,5-tetramethylimidazoline-1-oxyl-3-oxide radical-scavenging, 2,2′-azino-bis(3-ethylbenzo-thiazoline-6-sulfonic acid) radical-scavenging, and Fe^3+^-reducing activity. The DPPH^•^-scavenging reaction products of these assays were further analyzed by ultra-performance liquid chromatography coupled with electrospray ionization quadrupole time-of-flight tandem mass spectrometry (UPLC-ESI-Q-TOF-MS/MS) technology. In the five spectrophotometric assays, galangin and 3,5,7-trihydroxychromone dose-dependently increased their radical-scavenging (or Fe^3+^-reducing) percentages. However, galangin always gave lower IC_50_ values than those of 3,5,7-trihydroxychromone. In the UPLC-ESI-Q-TOF-MS/MS analysis, galangin yielded galangin-DPPH adduct MS peaks (*m*/*z* 662, 434, 301, 227,196, and 151) and galangin-galangin dimer MS peaks (*m*/*z* 538, 385, 268, 239, 211, 195, and 151). 3,5,7-Trihydroxychromone, however, only generated *m*/*z* 3,5,7-trihydroxychromone-DPPH adduct MS peaks (*m*/*z* 586, 539, 227, 196, and 136). In conclusion, both galangin and 3,5,7-trihydroxychromone could similarly undergo multiple antioxidant pathways, including redox-dependent pathways (such as electron transfer (ET) and ET *plus* proton transfer (PT)) and a non-redox-dependent radical adduct formation (RAF) pathway; thus, the null *B*-ring could hardly change their antioxidant pathways. However, it did improve their antioxidant levels in these pathways. Such improvement of the *B*-ring toward an antioxidant flavonol is associated with its π-π conjugation, which can provide more resonance forms and bonding sites.

## 1. Introduction

Flavonoids are important phenolic antioxidants derived from medicinal plants (especially those used in Chinese herbal medicines) [1,2]. Structurally, the scaffold of flavonoids consists of a chromone moiety (i.e., *A*/*C* fused ring) and a phenyl ring (i.e., *B*-ring), which are linked by a σ-bond with rotational possibilities. To the *A*/*C* fused ring or *B*-ring, more or less -OHs can be attached to construct phenolic -OHs [3]. The phenolic -OH, however, can occupy any of the 3-position, 5-position, 6-position, 7-position, 8-position, 2′-position, 3′-position, 4′-position, 5′-position, and 6′-position.

Particularly, when a phenolic -OH occupies at the 3-position, it is called a flavonol (Figure 1). Thus, flavonols are a subtype of the flavonoid family. Possibly owing to the importance of 3-OH, its antioxidant role has been analyzed by different chemical approaches. Quantum chemistry calculations have pointed out that the presence of a 3-OH along with a 2,3-double bond can facilitate an electron abstraction reaction in flavonols [4]. A combined experimental and theoretical study, however, suggested that the enthalpy change (ΔH_f_) has been calculated to be very low (32.28 kcal/mol); 3-OH, hence, is preferentially hydrogen abstracted to form a flavonol-3-O^•^ radical [5]. Undoubtedly, 3-OH is regarded as an important resource for the high antioxidant ability of flavonol.

In the last decades, there have also been 11 studies on the structure–activity relationship of phenolic -OH in flavonol [6,7,8,9,10,11,12,13,14,15,16]. In a word, the analysis of the structure–activity relationship of phenolic -OH (especially 3-OH) is relatively systematical. However, there has been no study discussing the role of flavonol scaffold (especially the *B*-ring) until now, to our knowledge.

To characterize the role of *B*-ring in antioxidant flavonols, galangin and its analogue 3,5,7-trihydroxychromone were selected as references in the study (Figure 2). As seen in Figure 2A, galangin bears a 3-OH and thus belongs to the flavonol group of flavonoids. However, when it loses the phenyl ring (i.e., the *B*-ring), it becomes its analogue 3,5,7-trihydroxychromone (Figure 2B). Thus, the comparison of the antioxidant activity between galangin and 3,5,7-trihydroxychromone can well characterize the role of *B*-ring in antioxidant flavonol.

To comparatively study the antioxidant abilities of galangin and 3,5,7-trihydroxychromone, we applied a set of antioxidant assays in this study. The antioxidant assays included ^•^O_2_^−^-scavenging assay (pH 7.4), 1,1-diphenyl-2-picrylhydrazyl radical (DPPH^•^)-scavenging assay, 2-phenyl-4,4,5,5-tetramethylimidazoline-1-oxyl 3-oxide radical (PTIO^•^)-scavenging assay (pH 7.4), 2,2′-azino-bis(3-ethylbenzo-thiazoline-6-sulfonic acid) radical ion (ABTS^+•^)-scavenging assay, and Fe^3+^-reducing assay (pH 3.6). These antioxidant assays were conducted by using a spectrophotometric method. The DPPH^•^-scavenging assay, however, was monitored by ultra-performance liquid chromatography coupled with electrospray ionization quadrupole time-of-flight tandem mass spectrometry (UPLC-ESI-Q-TOF-MS/MS) technology. From the perspective of antioxidant mechanism, however, these antioxidant assays are distinctive. On the other hand, their determining conditions are not fully identical. The DPPH^•^-scavenging assay and the ABTS^+•^-scavenging assay were performed in an organic solution, whereas PTIO^•^-scavenging assay, ^•^O_2_^−^-scavenging assay, and Fe^3+^-reducing assay were performed in an aqueous solution. To summarize, in the present study tried to use the above a set of antioxidant assays to compare the antioxidant abilities of galangin and 3,5,7-trihydroxychromone. We believe that the study will produce reliable experimental results concerning the role of *B*-ring in antioxidant flavonols.

## 2. Results and Discussion

The ^•^O_2_^−^ anion radical is an important radical of reactive oxygen species (ROS). The ^•^O_2_^−^ radical, however, can participate in a Haber-Weiss reaction, even in forming ^•^OH, a more reactive oxygen radical [17]. Both the ^•^O_2_^−^ and the ^•^OH radical, however, can cause oxidative DNA lesions (e.g., 8-oxo-7,8-dihydro-2′-deoxyguanosine lesion, 8-oxo-dG) that can induce a series of biological consequences, such as mutagenesis [18,19]. Thus, ^•^O_2_^−^ scavenging plays an important role in ROS scavenging (also termed as antioxidant action).

To test whether galangin and 3,5,7-trihydroxychromone can scavenge ^•^O_2_^−^, we performed a comparative analysis of the two using a pyrogallol auto-oxidation assay improved by our team [20]. The improved pyrogallol assay indicated that both galangin and 3,5,7-trihydroxychromone could concentration-dependently increase the ^•^O_2_^−^-scavenging percentages at pH 7.4 (Appendix A). It means that galangin and 3,5,7-trihydroxychromone have ^•^O_2_^−^-scavenging potential, which can be used to explain the beneficial effects of galangin [21,22].

However, ^•^O_2_^−^ scavenging has been involved in non-redox-dependent antioxidant pathway, such as radical adduct formation (RAF) [23,24,25,26], and in redox-dependent antioxidant pathways, including electron transfer (ET) [27] and ET *plus* proton transfer (PT) [24,25,28,29,30]. The so-called ET *plus* PT pathways are actually divided into several different subtypes, such as proton-coupled electron transfer (PCET) [29], double PT ET [28], ET-PT, and PT-ET [31].

To explore the non-redox-dependent RAF possibility, we mixed galangin and 3,5,7-trihydroxychromone, respectively, with a DPPH^•^ radical solution. Each of the reaction products was characterized by UPLC-ESI-Q-TOF-MS/MS analysis. As seen in Figure 3D, in the reaction product of galangin with DPPH^•^, at least four chromatographic peaks were found. Subsequent MS spectra determination suggested that four chromatographic peaks 1–4 similarly presented an *m*/*z* value of 662. This value, however, was exactly two less than the sum of the molecular weights (MW) of galangin (MW 270) and DPPH^•^ (MW 394). Thus, we initially assumed that the four peaks were galangin-DPPH adducts. Among the four peaks, peak 2 could be further broken down to produce the corresponding MS/MS spectrum, which showed four main fragments, namely, *m*/*z* 434, 301, 227, and 196 (Figure 3F). Of these, *m*/*z* 227 and 196 were presumed to be from the DPPH^•^ moiety (Figure 4A). In terms of these, the galangin-DPPH adduct was assumed as (I) in Figure 4A.

Moreover, as illustrated in Figure 3H, in the reaction product of galangin with DPPH^•^, one shoulder chromatographic peak was found to present an *m*/*z* of 538. This value is exactly two less than twice the molecular weight of galangin (MW 270). Thus, we initially assumed that two galangin molecules were dimerized via one covalent bond. The dimer was further broken in the MS/MS spectrum, which showed six main fragments, namely, *m*/*z* 385, 268, 239, 211, 195, and 151. Of these, three peaks (*m*/*z* 268, 239, and 211) were shared by the galangin molecule itself (Figure 3A). According to these and to the evidence from another flavonol, quercetin [32], the structure of galangin-galangin dimer was presumed as the (II) formula, which can be fully elucidated in Figure 4B.

It should be noted that the galangin-DPPH adduct and the galangin-galangin dimer may present other structural formulas. Even the (I) and (II) formulas also have other MS spectra elucidations. Nevertheless, it is certainly clear that, after the treatment with DPPH^•^, galangin could generate two main RAF products, i.e., galangin-DPPH adduct and galangin-galangin dimer. Thus, galangin could mediate the RAF pathway to exert its antioxidant action.

Similarly, 3,5,7-trihydroxychromone could also generate a RAF product (Figure 5). The product was identified as a 3,5,7-trihydroxychromone-DPPH adduct (III; Figure 6). However, there was no relevant peak in the 3,5,7-trihydroxychromone-3,5,7-trihydroxychromone dimer (Figure 5). In summary, both galangin and 3,5,7-trihydroxychromone could similarly undergo a RAF pathway to exert their antioxidant actions; however, their relative RAF possibilities were not identical. Galangin possessed more RAF possibilities than those of 3,5,7-trihydroxychromone because galangin could produce both a galangin-radical adduct and a galangin-galangin dimer, whereas 3,5,7-trihydroxy-chromone could only produce a 3,5,7-trihydroxychromone radical adduct.

In fact, the occurrence of RAF means that two antioxidant molecules, galangin and 3,5,7-trihydroxychromone, underwent redox-dependent antioxidant pathways to respectively transfer to two antioxidant radicals, i.e., galangin radical (C_15_H_9_O_5_^•^) and 3,5,7-trihydroxychromone radical (C_9_H_5_O_5_^•^). The reason is that RAF depends on the covalent bonding between two radicals [33]. If an antioxidant molecule has not been transformed into an antioxidant radical, RAF cannot take place. Therefore, the DPPH^•^-scavenging assay can also be utilized to explore the redox-dependent antioxidant pathways.

It has been documented that DPPH^•^-scavenging reaction is involved in several redox-dependent antioxidant pathways [34,35,36], including electron transfer (ET, also called electron abstraction [4,37]), ET plus PT (proton transfer or H^+^ transfer), hydrogen atom transfer (HAT [38], also called hydrogen abstraction [5]), and even RAF [39]. Essentially, HAT belongs to a special type of ET plus PT. As seen in Appendix A and Table 1, in the DPPH^•^-scavenging spectrophotometer assay, both galangin and 3,5,7-trihydroxychromone efficiently scavenged a DPPH^•^ radical. This preliminarily suggested that their antioxidant action was also involved in the redox-dependent antioxidant pathways.

Like in case of the DPPH^•^-scavenging assay, our team believes that the PTIO^•^-scavenging assay was also involved in the redox-dependent antioxidant pathways. However, the PTIO^•^-scavenging assay was conducted in an aqueous solution [40]. As seen in Appendix A and Table 1, the PTIO^•^-scavenging actions of galangin and 3,5,7-trihydroxychromone resembled their DPPH^•^-scavenging actions. It can be inferred that both galangin and 3,5,7-trihydroxychromone could also have redox-dependent antioxidant pathways in both the aqueous media and the organic media.

As mentioned above, a typical redox-dependent antioxidant pathway is mediated by at least an ET reaction. To test the ET possibility, we performed a comparative measurement of both galangin and 3,5,7-trihydroxychromone using an Fe^3+^-reducing power spectrophotometer assay at pH 3.6. In the Fe^3+^-reducing power assay, the acidic condition (pH 3.6) suppressed the H^+^ ionization (i.e., PT) [41]; thus, the Fe^3+^-reducing power assay was recognized as an ET reaction. The two, however, could increase their Fe^3+^-reducing percentages in a dose dependent manner (Appendix A). This indicates that they have ET potentials. Such ET potentials can be partly supported by the evidence from the ABTS^•+^-scavenging spectrophotometer assay. The ABTS^•+^-scavenging assay was mainly characterized by ET because the ABTS^•+^ generation depended on the ET from (NH_4_)_2_ABTS [42,43]. Both galangin and 3,5,7-trihydroxychromone, however, presented a strong ABTS^•+^ -scavenging ability at 0–10 μg/mL (Appendix A and Table 1).

Now it is clear that both galangin and 3,5,7-trihydroxychromone could similarly undergo multiple antioxidant pathways, including redox-dependent pathways (such as ET and ET plus PT) and a non-redox-dependent RAF pathway. However, in the redox-dependent or non-redox-dependent antioxidant assays, galangin always gave lower IC_50_ values than those of 3,5,7-trihydroxychromone (Table 1). This means that the null *B*-ring could hardly change their antioxidant pathways but could improve their antioxidant levels in these pathways. Such improvement of the *B*-ring toward an antioxidant flavonol can be definitively attributed to the presence of the *B*-ring because the null *B*-ring is the mere difference between galangin and 3,5,7-trihydroxychromone.

As shown in Figure 1 and Figure 2, the *B*-ring is linked to the chromone moiety via a σ-bond. Although the σ-bond can rotate freely, its preferential conformation suggests that the *B*-ring shares a plane with the chromone moiety. This provides a π-π conjugation possibility between a *B*-ring and a chromone moiety. However, our latest study suggested that π-π conjugation can improve the antioxidant levels of phenolic antioxidants [24] because it can provide more resonance forms.

As mentioned above, 3-OH plays a critical role in antioxidant flavonol. Therefore, given that 3-OH loses a hydrogen atom via redox-dependent pathways, it may be transformed into a 3-*O*^•^ radical. Correspondingly, the galangin molecule and the 3,5,7-trihydroxychromone molecule, respectively gave rise to the galangin-3-*O*^•^ radical (C_15_H_9_O_5_^•^) and the 3,5,7-trihydroxychromone-3-*O*^•^ radical (C_9_H_5_O_5_^•^) (Figure 7). However, in the galangin-3-*O*^•^ radical, there were five resonance formulas (Figure 7A), whereas there were only two resonance formulas in the 3,5,7-trihydroxychromone-3-*O*^•^ radical. More resonance formulas mean that the galangin-3-*O*^•^ radical was more stable than the 3,5,7-trihydroxychromone-3-*O*^•^ radical. The higher stability of an antioxidant radical intermediate, however, indicates a higher ability of an antioxidant molecule, and, thus, the null *B*-ring improved the antioxidant level of flavonol via π-π conjugation.

More resonance forms also suggest more bonding sites. This can be used to explain the above findings that: (*i*) galangin produced more adducts with DPPH^•^ in the chromatographic peaks than those of 3,5,7-trihydroxychromone (Figure 3D and Figure 5N) and (*ii*) galangin produced at least one dimeric peak in Figure 3H; in contrast, 3,5,7-trihydroxychromone produced no relevant peaks. Especially, if the resonance formula (IV) was combined with a galangin-3-*O*^•^ radical (Figure 7A), the bonding sites would be between 3-*O* and 2″-*C*. Such linkage led to a galangin-galangin dimer (II) in Figure 4B. In a word, π-π conjugation could provide not only more resonance formulas to stabilize the antioxidant-radical intermediates but also more bonding sites to generate RAF products.

## 3. Materials and Methods

### 3.1. Chemicals

Galangin (C_15_H_10_O_5_, CAS number 548-83-4, MW 270.2, purity 98%) and 3,5,7-trihydroxychromone (C_9_H_6_O_5_, CAS number 31721-95-6, MW 194.1, purity 98%) were obtained from BioBioPha Co., Ltd. (Kunming, China). Pyrogallol, 2,4,6-tripyridyl triazine (TPTZ), and (±)-6-hydroxyl-2,5,7,8-tetramethylchromane-2-carboxylic acid (Trolox) were obtained from Sigma-Aldrich (Shanghai, China). 1,1-Diphenyl-2-picrylhydrazyl radical (DPPH^•^, C_18_H_12_N_5_O_6_) was obtained from Aladdin Chemical, Ltd. (Shanghai, China). The 2-phenyl-4,4,5,5-tetramethylimidazoline-1-oxyl-3-oxide radical (PTIO^•^) was obtained from TCI Chemical Co. (Shanghai, China). (NH_4_)_2_ABTS [2,2′-azino-bis (3-ethylbenzo-thiazoline-6-sulfonic acid diammonium salt)] was obtained from Amresco Chemical Co. (Solon, OH, USA). Tris-hydroxymethyl aminomethane (Tris) was obtained from Dingguo Biotechnology, Ltd. (Beijing, China). Water and acetonitrile were of HPLC grade. FeCl_3_·6H_2_O and the other reagents were of analytical grade and purchased from Guangdong Guanghua Chemical Plants Co., Ltd. (Shantou, China).

### 3.2. Superoxide Anion (^•^O_2_^−^)-Scavenging Spectrophotometer Assay (Pyrogallol Autoxidation Method)

The superoxide anion (^•^O_2_^−^)-scavenging activity was determined using a method previously developed in our laboratory [20]. Briefly, a 10- to 50-μL sample solution (0.5 mg/mL) was added to 0.05 M of Tris-HCl methanol/water (1/4, *v*/*v*) buffer (pH 7.4) containing Na_2_EDTA (1 mM) and the total volume was adjusted to 980 μL using the buffer. Twenty microliters of pyrogallol (1,2,3-trihydroxylbenzene) solution (60 mM in 1 mM of HCl) was added to the sample, and the resulting mixture was vigorously agitated before being analyzed at 325 nm every 30 s for 5 min. The ^•^O_2_^−^ radical-scavenging ability was calculated as follows:$$Inhibition\%=\frac{\left(\frac{\Delta A_{325\;\mathrm{nm},control}}{T}\right)-\left(\frac{\Delta A_{325\;\mathrm{nm},sample}}{T}\right)}{\left(\frac{\Delta A_{325\mathrm{nm},control}}{T}\right)}\times 100\%,$$
where *ΔA*_325 nm, *control*_ is the increase in the *A*_325 nm_ value of the mixture without the sample, *ΔA*_325 nm, *sample*_ is the increase in the *A*_325 nm_ value of the mixture with the sample, and *T* is the time required for the determination (5 min in this case).

### 3.3. UPLC-ESI-Q-TOF-MS/MS Analysis of DPPH^•^ Reaction Products with Galangin and 3,5,7-Trihydroxychromone

The reaction of galangin with 3,5,7-trihydroxychromone proceeded under the conditions described in our previous study [44]. In brief, a methanol solution of galangin was mixed with a methanol DPPH^•^ solution at a molar ratio of 1:2, and the resulting mixture was incubated for 10 h at room temperature. The product was then filtered through a 0.22-μm filter for UPLC-ESI-Q-TOF-MS/MS analysis.

The UPLC-ESI-Q-TOF-MS/MS analysis was based on the method described in our previous study [45]. The UPLC-ESI-Q-TOF-MS/MS analysis system was equipped with a C_18_ column (2.0 mm i.d. × 100 mm, 2.2 μm, Shimadzu Co., Kyoto, Japan). The mobile phase was used for the elution of the system and consisted of a mixture of acetonitrile (phase A) and 0.1% formic acid water (phase B). The column was eluted at a flow rate of 0.2 mL/min with the following gradient elution program: 0–2 min, maintained at 30% B; 2–10 min, 30–0% B; and 10–12 min, 0–30% B. The sample injection volume was set at 1 μL for the separation of the different components. Q-TOF-MS/MS analysis was performed on a Triple TOF 5600*^plus^* mass spectrometer (AB SCIEX, Framingham, MA, USA) equipped with an ESI source, which was run in the negative ionization mode. The scan range was set at 100–2000 Da. The system was run with the following parameters: ion spray voltage, −4500 V; ion source heater, 550 °C; curtain gas (CUR, N_2_), 30 psi; nebulizing gas (GS1, air), 50 psi; and Tis gas (GS2, air), 50 psi. The declustering potential (DP) was set at −100 V, whereas the collision energy (CE) was set at −40 V with a collision energy spread (CES) of 20 V. The RAF final products were quantified by extracting the corresponding formula from the total ion chromatogram and integrating the corresponding peak. The above experiments were repeated using 3,5,7-trihydroxychromone.

### 3.4. DPPH^•^-Scavenging Spectrophotometer Assay

The DPPH^•^ radical-scavenging assay was conducted according to previously reported procedures from the literature [46]. The experimental protocols, experimental apparatus, and formula for calculating the inhibition percentages were similar to those previously reported. In contrast to this previous report, the samples tested in this study were galangin and 3,5,7-trihydroxychromone, with Trolox being used as the positive controls. The IC_50_ values of galangin and 3,5,7-trihydroxychromone are shown in Table 1.

### 3.5. PTIO^•^-Scavenging Spectrophotometer Assay

The PTIO^•^-scavenging spectrophotometer assay was conducted in accordance with our method [40]. In brief, the test sample solution (x = 0–10 μL, 0.5 mg/mL) was added to (20 − x) μL of methanol, followed by 80 μL of an aqueous PTIO^•^ solution. The aqueous PTIO^•^ solution was prepared using 0.1 mM of phosphate buffer/methanol (1/4, *v*/*v*) solution (pH 7.4). The mixture was maintained at 37 °C for 2 h, and the absorbance was then measured at 560 nm using a microplate reader (Multiskan FC, Thermo Scientific, Shanghai, China). The PTIO^•^ inhibition percentage was calculated as follows:$$Inhibition\;\%=\frac{A_{0}-A}{A_{0}}\times 100\%$$
where *A*_0_ is the absorbance of the control without the sample and *A* is the absorbance of the reaction mixture with the sample.

### 3.6. Fe^3+^-Reducing Antioxidant Spectrophotometer Assay

The Fe^3+^-reducing antioxidant spectrophotometer assay used in this study was adapted from the method reported by Benzie and Strain [41]. This assay can be used to give an indication of the reducing ability of a material or mixture. The assay was performed in a buffer with a pH of 3.6. Briefly, according to a ratio of 1:1:10, the determining reagent was freshly prepared by mixing together 10 mM of TPTZ and 20 mM of FeCl_3_ in 0.25 M of acetic acid–sodium acetate/methanol (1/4, *v*/*v*) buffer (pH 3.6). The test sample (*x* = 0–10 μL, 0.1 mg/mL) was added to (20 − *x*) μL of methanol, followed by 80 μL of a determining reagent. The absorbance was read at 593 nm after 30 min of incubation at 37 °C against a blank consisting of an acetate buffer. The relative reducing power of the sample compared with the maximum absorbance was calculated using the following formula:$$Relative\;reducing\;power\%=\frac{A-A_{min}}{A_{max}-A_{min}}\times 100\%,$$
where *A_max_* is the maximum absorbance, *A_min_* is the minimum absorbance, and *A* is the absorbance of the sample.

### 3.7. ABTS^•^-Scavenging Spectrophotometer Assay

The ABTS^+•^-scavenging activity was estimated using the method by Li et al. [47]. The ABTS^+•^ was produced by mixing 350 μL of (NH_4_)_2_ABTS (7.4 mM) with 350 μL of K_2_S_2_O_8_ (2.6 mM). The mixture was kept in the dark at room temperature for 12 h to produce an ABTS^+•^ aqueous solution. The aqueous solution was then diluted with methanol (about 1:50) to its A_734 nm_ value of 0.30 ± 0.02. To estimate the ABTS^+•^-scavenging activity, we added the test sample (x = 0–10 μL, 0.1 mg/mL) to (20 − x) μL of methanol, followed by 80 μL of the ABTS^+•^ diluted solution, and then the absorbance at 734 nm was measured 6 min after the initial mixing using methanol as the blank. The percentage inhibition was calculated according to the formula presented in Section 3.5.

### 3.8. Statistical Analysis

The results were reported as the mean ± SD of three independent measurements, the IC_50_ values were calculated by linear regression analysis, and independent-sample *T* tests were performed to compare the different groups. A *p* value of less than 0.05 was considered statistically significant. Statistical analyses were performed using the software SPSS for Windows version 17.0 (SPSS Inc., Chicago, IL, USA). All of the linear regression analyses described in this paper were processed using Origin 2017 professional software (OriginLab, Northampton, MA, USA).

## 4. Conclusions

Galangin with a null *B*-ring and 3,5,7-trihydroxychromone without a *B*-ring could similarly pass through redox-dependent pathways (such as ET, ET plus PT, and HAT) and a non-redox-dependent RAF pathway to show an antioxidant action. Although the null *B*-ring does not change the antioxidant pathways, however it improves the antioxidant levels in these pathways. Such improvement of the *B*-ring can be attributed to its π-π conjugation with chromone moiety. The π-π conjugation can provide not only more resonance forms but also more bonding sites.

## Figures and Tables

**Figure 1 molecules-23-03083-f001:**
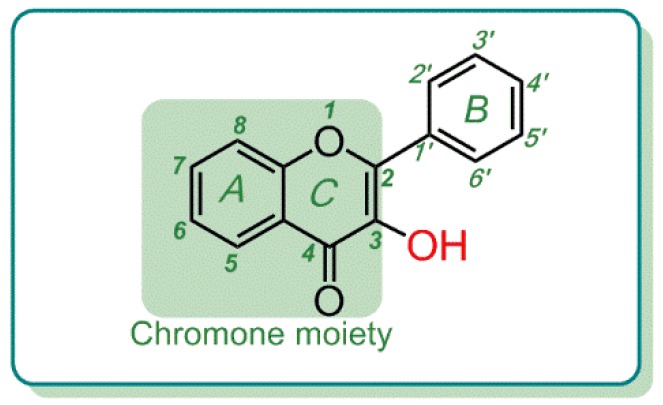
Scaffold of flavonol.

**Figure 2 molecules-23-03083-f002:**
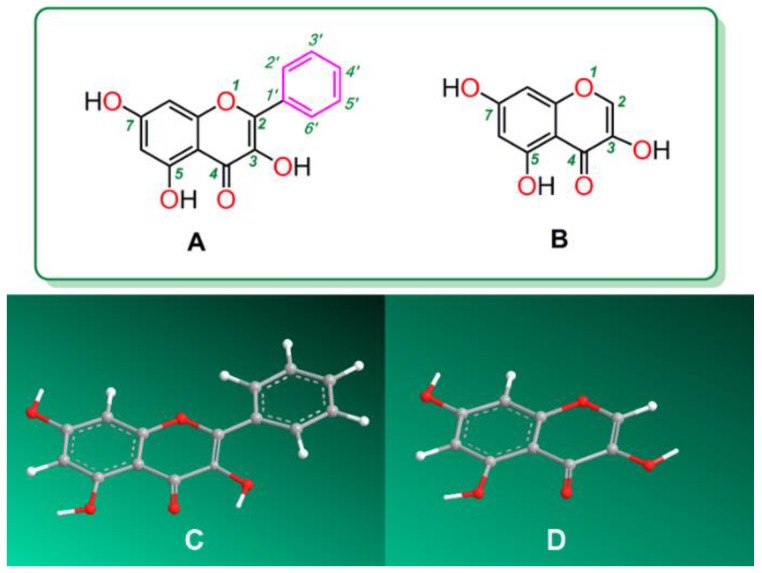
Structural formula and molecular models of galangin and 3,5,7-trihydroxychromone: (**A**), structural formula of galangin; (**B**)**,** structural formula of 3,5,7-trihydroxychromone; (**C**)**,** preferential conformation-based molecular model of galangin; and (**D**), preferential conformation-based molecular model of 3,5,7-trihydroxychromone. The molecular models were created using Chem3D Pro 14.0.

**Figure 3 molecules-23-03083-f003:**
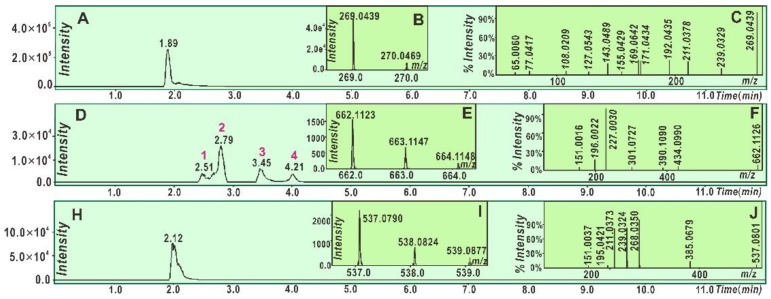
Main results of galangin in the UPLC−ESI−Q−TOF−MS/MS analysis: (**A**) total ion chromatographic diagram of galangin; (**B**) primary MS spectra of galangin; (**C**) MS/MS spectra of galangin; (**D**) total ion chromatographic diagram of the RAF products of galangin with DPPH^•^; (**E**) primary MS spectra of galangin-DPPH^•^ adduct; (**F**) MS/MS spectra of galangin-DPPH^•^ adduct; (**H**) total ion chromatographic diagram of the possible dimeric products of galangin; (**I**) primary MS spectra of the galangin-galangin dimer; and (**J**) MS/MS spectra of the galangin-galangin dimer.

**Figure 4 molecules-23-03083-f004:**
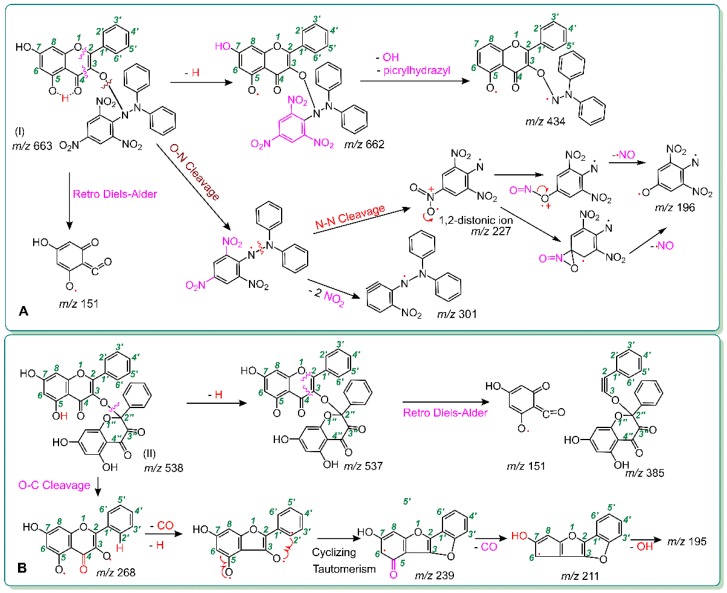
Proposed RAF products of the galangin reaction with DPPH and their MS elucidations. (**A**) For the galangin-DPPH adduct and the MS elucidation; (**B**) for the galangin-galangin dimer and the MS elucidation (The MS spectra were in the negative ion model, and the charge imposed by the MS field was not marked. Other linking sites between two galangin moieties or between a galangin moiety and a DPPH moiety should not be excluded. Other reasonable cleavages should not be excluded in the MS elucidation).

**Figure 5 molecules-23-03083-f005:**
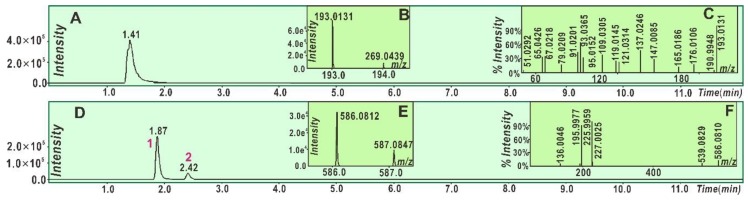
Main results of 3,5,7-trihydroxychromone in the UPLC-ESI-Q-TOF-MS/MS analysis: (**A**) Total ion chromatographic diagram of 3,5,7-trihydroxychromone; (**B**) primary MS spectra of 3,5,7-trihydroxychromone; (**C**) MS/MS spectra of 3,5,7-trihydroxychromone; (**D**) total ion chromatographic diagram of 3,5,7-trihydroxychromone-DPPH adduct; (**E**) primary MS spectra of 3,5,7-trihydroxychromone-DPPH^•^; and (**F**) MS/MS spectra of 3,5,7-trihydroxychromone-DPPH adduct.

**Figure 6 molecules-23-03083-f006:**
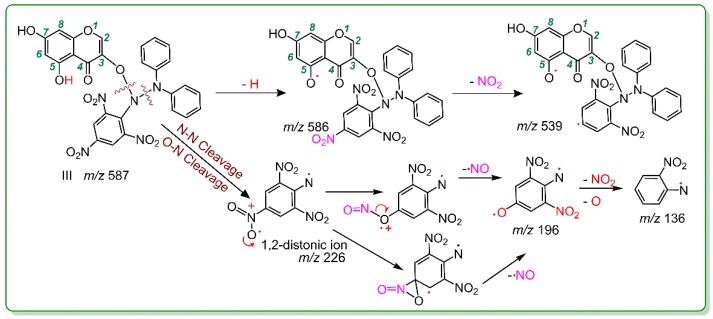
Proposed RAF product of 3,5,7-trihydroxychromone-DPPH adduct and its MS elucidation (Other linking sites between a 3,5,7-trihydroxychromone moiety and a DPPH moiety should not be excluded. Other reasonable cleavages should not be excluded in the MS elucidation.).

**Figure 7 molecules-23-03083-f007:**
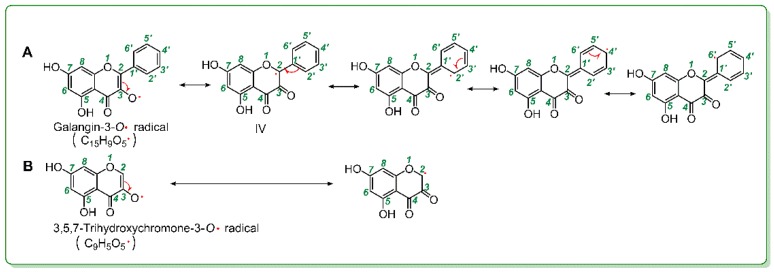
Resonance formula of the galangin-3-*O*^•^ radical (C_15_H_9_O_5_^•^, **A**) and the 3,5,7-trihydroxychromone-3-*O*^•^ radical (C_9_H_5_O_5_^•^, **B**).

**Table 1 molecules-23-03083-t001:** IC_50_ values (µM) of galangin and 3,5,7-trihydroxychromone in five antioxidant spectrophotometer assays.

Assays	Galangin	3,5,7-Trihydroxychromone	Trolox
^•^O_2_^−^ scavenging	108.7 ± 4.4 ^a^	125.4 ± 4.9 ^b^	4968.3 ± 157.8 ^c^
DPPH^•^ scavenging	10.2 ± 0.3 ^a^	134.1 ± 21.4 ^c^	36.9 ± 1.0 ^b^
PTIO^•^ scavenging	176.7 ± 24.3 ^b^	395.1 ± 33.2 ^c^	186.3 ± 7.2 ^a^
Fe^3+^ reducing	22.9 ± 1.3 ^a^	42.5 ± 1.6 ^b^	21.5 ± 0.8 ^a^
ABTS^+•^ scavenging	19.7 ± 0.1 ^a^	20.9 ± 0.3 ^b^	43.2 ± 1.7 ^c^

The IC_50_ value (in micromolar unit) was defined as the final concentration of 50% radical inhibition or relative reducing power, calculated by linear regression analysis, and expressed as the mean ± SD (*n* = 3). The linear regression was analyzed using the Origin 2017 Professional software. IC_50_ values with different superscripts (a, b, or c) in the same row were significantly different (*p* < 0.05). Trolox was the positive control. The dose–response curves are listed in Appendix A.

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
