# Peer review of "A Null *B*-Ring Improves the Antioxidant Levels of Flavonol: A Comparative Study between Galangin and 3,5,7-Trihydroxychromone"

_molecules, 2018, doi:10.3390/molecules23123083_

Round 1
Reviewer 1 Report
The manuscript “Null B-Ring Improves the Antioxidant Levels of Flavonol: A Comparative Study between Galangin and 3,5,7-Trihydroxychromone” by Xiaojian Ouyang shows a solid and well done experimental work. The methodology is correct and appropriate, and the results are consistent. The aims of the paper is clearly defined. The introduction can be considered updated, and the authors addressed well the subject studied. The materials and methods are clearly defined and allow a perspective of accurracy/precision of data obtained by the authors. The discussion is supported by data. The conclusion is concise and objective. The paper, according to my perspective deserves publication after these revisions:
1) English spell-check is advised: there is some awkward English usage and I strongly suggest a good reading by an English editor. For example Page 3 lines 107 to 108: the reaction products was not obtained by UPLC-ESI-Q-TOF-MS/MS, they were characterized. Others as well.
2) Page 3, line 98: it is not clear what the authors refer to with suppl. 2 (do they mean Fig. S1 or Tab. S1?)
3) Page 3 line 102: the abbreviation RAF must be explicit as was done for ET in the same line
4) Line 117: Fig. 4N does not exist: what figure do the authors refer to?
Reviewer 2 Report
The paper required an improved presentation of data - through improved figures.
Reviewer 3 Report
The results are interesting. The study is well done but the presentation is complicated in my opinion. There are a lot of formulas and numbers within the paragraphs. Auhtors should think in another more friendly presentation for the reader. Moreover, they shoud enphasize the main conclusion of this investigation.
The results are not very original since there are a lot of reports with these experiments.
Reviewer 4 Report
Review on manuscript:
Null B-Ring Improves the Antioxidant Levels of Flavonol: A Comparative Study between Galangin and 3,5,7-Trihydroxychromone
by Xiaojian Ouyang, Xican Li, Wenbiao Lu, Xiaojun Zhao and Dongfeng Chen
submitted to Molecules
In the manuscript submitted for comments the Authors studied role of B-ring in antioxidant properties of flavonols, by comparison between activity of galangin with a null B-ring and 3,5,7-trihydroxychromone without a B-ring using different spectrophotometric assays.
Generally the manuscript is interesting and prepared correctly. So, after minor revision could be accepted for publication in Molecules journal.
Detailed recommendations:
lines 68-82 – literature review should ended with a clearly formulated goal of the research undertaken, in its present form it looks more like a summary, so this part should be rewritten,
line 79 – did the authors really use the Cu2+ reducing assay?
lines 202-203 – this information should be placed under the table as its legend,
lines 295-296 – table 1 contains IC50 values,
References – double numbering of the cited literature should be removed.
